# Towards AI as a Collaborative Partner: A Taxonomy of AI Agent Behavior in Software Engineering

## Abstract

The ongoing transition of Large Language Models in software engineering from one-shot code generators into agentic partners requires a shift in how we define and measure success. While models are becoming more capable, the industry lacks a clear understanding of the behavioral norms that make an interactive SWE agent effective in collaborative software development in the enterprise. This work addresses this gap by presenting a taxonomy of desirable SWE agent behaviors, synthesized from 91 sets of developer-defined rules for SWE agents and validated through interviewing 15 experienced professional developers. In this taxonomy, we identify four core expectations: *Adhere to Standards and Processes*, *Ensure Code Quality and Reliability*, *Solve Problems Effectively*, and *Collaborate with the Developer*. These findings offer a concrete vocabulary for aligning SWE agent behavior with developer preferences, enabling researchers and practitioners to move beyond correctness-only benchmarks and start designing evaluations that reflect the socio-technical nature of professional software development in enterprises.

## CCS Concepts

• **Human-centered computing** → **Empirical studies in HCI**.

## Keywords

Human-AI Collaboration, Software Engineering, AI Agents, LLM Evaluation

**ACM Reference Format:**

Anonymous Author(s). 2025. Towards AI as a Collaborative Partner: A Taxonomy of AI Agent Behavior in Software Engineering. In *Proceedings of Make sure to enter the correct conference title from your rights confirmation email (Conference acronym 'XX)*. ACM, New York, NY, USA, 9 pages. https://doi.org/XXXXXXX.XXXXXXX

## 1 Introduction

Powered by recent advancements in Large Language Models (LLMs), the nature of generative AI tools for software engineering has been undergoing a rapid transformation over the past few years, shifting from passive code completers [27] to increasingly autonomous agents capable of tackling complex tasks including testing [12], migrations [20], bug fixing [18, 22], and more. Many SWE agents are also designed to be interactive, and their wide adoption is establishing deep human-AI collaboration as an industry norm. This evolution marks a critical paradigm shift: from AI as a tool to be wielded by a developer to AI as a collaborative partner integrated into a software team's workflow [13, 16].

As agents assume this role of "partner," the definition of success becomes increasingly ambiguous. While current models are becoming more capable of generating correct code, the industry lacks a clear understanding of the behavioral norms that make a SWE agent an effective contributor within a software development team. Unlike a simple tool, an AI partner must demonstrate sound judgment, adhere to team values and norms, and communicate effectively - qualities that are often implicit and difficult to define.

This ambiguity creates a significant gap in how we measure progress. The field has been heavily guided by benchmarks like SWE-bench [10] and LiveCodeBench [8], which define success primarily in terms of functional correctness. This focus on just the functional correctness leaves us unequipped to assess the agent's collaborative behaviors and process adherence-capabilities that are essential to creating a high-quality user experience and establishing user trust.

To address this challenge and move beyond a monolithic view of agent performance, we ask the overarching research question: *What are the core behaviors an intelligent agent should be evaluated for in software engineering?*

To answer this question, we took a two phased approach. In the first phase, we derived a taxonomy of four core behavioral expectations and 15 specific attributes through a qualitative analysis of 91 real-world agent rule sets deployed in a large technology company. Agent rules are often defined in project-level configuration files[1], such as CLAUDE.md or GEMINI.md, that allow developers to provide custom instructions and behavioral guidelines to steer how an AI agent approach tasks within a given codebase.

As these rules are heavily influenced by the user's experience with the current system, they represent an evolving baseline rather than a fixed standard. Therefore, we then triangulated these findings in the second phase, by conducting 15 semi-structured interviews with senior developers in the same company. In these interviews, we framed the conversation around the values and mindsets that are embodied by successful software engineers. Participants then reflected on how junior software engineers and AI agents measure up against these standards. This framing allows us to define what a true collaborative partner looks like, unconstrained by current abilities of technology and prescribe the north-star for values and behaviors an agent needs to exhibit beyond just successful task completion.

By characterizing these behavioral expectations of AI agents as collaborative partners, we provide the missing vocabulary needed to close the current evaluation gap, enabling the field to move beyond

---

[1]An introduction to agent rule files: https://cursor.com/docs/context/rules

functional correctness and start assessing an agent's capacity for effective teamwork.

## 2 Related Work

The optimization targets of AI in software engineering have been heavily expressed in and codified by benchmarks. Early benchmarks such as HumanEval [5] and MBPP [1] established functional correctness as the primary metric, treating the agent as a black-box code generator. Success was measured by pass@k rates on unit tests, a suitable method for evaluating single-shot code completion. As tasks grew more complex, benchmarks like SWE-Bench [10] and SWE-Lancer [19] maintained this focus on final outcomes, albeit for more challenging, multi-file problems. Most recently, with the rise of autonomous agents in coding, the research focus has shifted from the final output to the process of generating it. Frameworks like MAST [21] and TRAIL [6] provide valuable perspectives and common failure modes to look out for.

One important capability of coding agents that many existing evaluation techniques do not assess is the quality of the agent's interactions with the developer. In contrast, in domains such as medicine and education, human-centric metrics have been adopted when assessing LLMs' capabilities. For instance, studies in medicine evaluate AI not just for diagnostic accuracy but also for the quality and empathy of its communication with patients [2, 15]. Similarly, in education, AI tutors are assessed on their ability to provide scaffolding and encouragement, not just correct answers [11, 17, 25].

Within the AI for SWE domain, researchers are beginning to acknowledge this need. In an observational study, Kumar et al. [13] found a direct correlation between successful task completion and the amount of human-agent communication, highlighting the importance of interaction quality. However, the specific characteristics that constitute high-quality human-AI interaction in this domain remain unclear.

One source of developers' expectations for agent behavior is agent rules, a mechanism for developers to provide project-level context and directives that has recently seen growing adoption across the industry. In a study by Jiang et al. [9], 36% of repos studied have agent rule files that contain behavior-related guidelines. Similarly, in Chatlatanagulchai et al.'s study [4], 24.4% of agent context files contained specific instructions on the desired behavior and roles of agentic coding, and methods for integrating other AI tools .

While these studies take a descriptive approach to analyzing the content of agent rules in open source software projects, our work leverages data from one of the largest enterprise codebases, and focuses primarily on the expected behaviors expressed in these rules to establish a normative taxonomy for the evaluation of agent behavior.

One confounding factor in establishing a forward looking taxonomy of agent behaviors from agent rules is that these rules could reflect current state of the technology rather than the ideal expectations of a truly collaborative SWE partner. To account for this, we draw insights from studies that focus on the attributes of great software engineers.

Li et.al [14] conducted mixed-methods research with experienced software engineers at Microsoft to understand attributes that make a great software engineer. They identified 53 attributes spanning themes like being curious, long-term thinking, making well-considered and informed decisions, sharing knowledge with others, and continuously learning and improving.

In more specific domains and contexts, Hewner [7] explored desired qualifications for new game developers, identifying interpersonal skills—such as the ability to collaborate effectively and set aside personal ego—as crucial alongside core technical competencies. Similarly, Begel et al. [3] observed that in pair programming settings, participants favored partners who offered complementary perspectives, demonstrated open-mindedness, and possessed strong communication skills.

These studies clearly demonstrate that being an effective software engineer in a team goes beyond just technical competency. In our interviews with senior developers, we build on this literature, by explicitly asking participants to compare their expectations for an AI agent to those for a junior human developer.

## 3 Agent Rule Analysis

Building on these bodies of work, our initial effort focused on creating a taxonomy of desirable agent behaviors immediately applicable to an enterprise software engineering context. To identify these behaviors, we performed a qualitative content analysis of user-defined rules for steering the behavior of a coding agent used at a global technology company. These rules, stored in the codebase as a markdown file, customize the agent's default behavior as a preamble of user prompts.

### 3.1 Methods

To create a taxonomy of desirable agent behaviors in enterprise software engineering, we analyzed 91 project-level rule files for coding agents at a global technology company. Collected in late July 2025, about a month after support for agent rules was introduced to software developers at the company, this dataset represents perspectives from early adopters of agent rules in this organization.

Two of the authors performed iterative open coding on 15 rule files, which led to a codebook consisting of 15 behavioral attributes grouped by four themes. To scale the analysis, we employed an LLM-based annotator to code the remaining corpus. We instructed the model to first segment each file into a set of coherent rules and then annotate each rule with up to three codes from the codebook. On a validation set of 5 files (95 rules), the annotator achieved 94.4% precision and 91.2% recall with zero hallucinations, a performance we deemed acceptable for our qualitative analysis.

### 3.2 Findings

Our analysis of the agent rules defined by software engineers reveals four core expectations and 15 specific behaviors for AI agents, summarized in Table 1. In the rest of this section, we describe these expectations in detail. We redacted proprietary information with descriptive labels enclosed in angle brackets.

*3.2.1 Adhering to Standards and Processes.* A predominant expectation was that the agent must strictly adhere to established standards and project-specific processes. This expectation highlights

**Table 1: Human-Centered Taxonomy of AI Agent Behavior: Core Expectations, Behaviors, and Example Rules. % Files represents the percentage of files in our dataset that contained rules pertaining to each behavior.**

| Expectation | Behavior | Example Agent Rule from Corpus | % Files |
|---|---|---|---|
| **1. Adhere to Standards and Processes** | Following Established Best Practices | "Follow the [Angular Style Guide](<URL>) for more details" | 45.05 |
| | Following Project Workflows and Conventions | "After creating your new _test.go file, add a go_test target to the test/<package> file." | 92.31 |
| **2. Ensure Code Quality and Reliability** | Maintain Code Style | "Follow Local Style: Adhere to existing patterns and naming conventions... This takes precedence over general style guides." | 57.14 |
| | Write Readable and Maintainable Code | "Write for Others: Code should be written with the assumption that someone else will read, understand, and maintain it." | 36.26 |
| | Build Robust and Performant Software | "Unsubscribe from observables using take(1) or other lifecycle-aware operators to prevent memory leaks." | 46.15 |
| **3. Solve Problems Effectively** | Understand Project Context before Acting | "Before creating the first SQL query, *you must* read the '.proto' files linked from the <path>/sql.md to understand the fields that are very likely to come up in queries." | 90.11 |
| | Work Incrementally and Iteratively | "Incremental Changes: Make the smallest possible code change, then run tests. Fix failures before making further changes." | 11.00 |
| | Validate Work Proactively | "After finishing any changes, always run our unit tests by running <command> and iterating until these tests pass." | 37.36 |
| | Maintain Task Focus | "Defer Unrelated Tasks: If you identify a necessary but out-of-scope task (e.g., a needed data model refactoring), leave a TODO comment." | 9.89 |
| | Infer Intent from Context | "If the user does not write any test description, the AI agent should still try inferencing the test case from the name." | 19.78 |
| | Learn by Example | "Inspect other tools in the tools/ dir and copy their approaches." | 32.97 |
| **4. Collaborate with the Developer** | Communicate Effectively | "STRICTLY FORBIDDEN from starting messages with 'Great' or 'Certainly'... It is important you be clear and technical in your messages." | 30.77 |
| | Seek Help and Clarification | "You are allowed to ask the user questions. Be sure to use a clear and concise question that will help you move forward with the task." | 25.27 |
| | Plan Collaboratively and Analyze Trade-offs | "Critically evaluate all requests. If a prompt is ambiguous, please challenge it and propose a better alternative, explaining the trade-offs..." | 8.79 |
| | Learn from Feedback and Past Experiences | "At the end of every task, or upon making an error, you MUST update the lessons_learned.md file...This process is essential for self-correction and knowledge retention." | 14.29 |

the "brownfield" nature of software development in a large enterprise where AI-generated code is expected to integrate seamlessly into a mature, existing codebase.

Developers frequently instructed the agent to "follow," "adhere to," "consult," and "use" pre-existing, widely-accepted guidelines for specific programming languages and frameworks within the company. These included documents like "CSS Best Practices," "efficient java guide," "Go Style," and the "TypeScript style guide." The specificity of these rules can vary across projects. While some rules simply link to online documentation, others underscore key steps directly in the rules file itself, indicating differing degrees of confidence in the agent's capacity to identify and implement relevant guidelines.

Beyond general best practices, developers specified a multitude of *project-specific* procedures and conventions. The vast majority

(92.31%) of files in our dataset contained guidelines related to "Following Project Workflows and Conventions". These rules governed local practices such as dependency management, interaction with external systems, test generation and execution, API usage patterns, naming conventions, and code organization. By providing these rules, developers effectively mandated an "orientation" for the AI agent to ensure the agent's contributions are not only functional but also conformant.

*3.2.2 Ensure Code Quality and Reliability.* Our analysis of agent rules highlights developers' deep concern for code quality, extending far beyond functional correctness. Developers instructed the agent to produce code consistent in style, maintainable, reliable, and performant.

First, a significant portion of the rules centered on enforcing stylistic consistency, a critical factor for maintaining a large-scale, multi-author codebase. These instructions fell into three categories:

- **Guideline Application:** Developers frequently directed the agent to follow links of official style guides. These references were often supplemented with specific rules for naming conventions, import ordering, or syntax.
- **Tool Execution:** Adherence was often automated by instructing the agent to execute standard formatting and linting tools as part of its workflow.
- **Contextual Adaptation:** Some developers expected the agent to infer and adopt the style of the surrounding code and instructed the agent to prioritize local conventions over global ones when they are in conflict.

Second, developers emphasized that AI-generated code should be easy for humans to understand, modify, and debug. This expectation frames code not as a set of instructions for a machine, but as a form of communication within a development team. Specifically, developers provided several types of guidance about code maintainability:

- **High-Level Principles:** Adherence to established software engineering philosophies like DRY ("Don't Repeat Yourself"), "Loose Coupling," and the "Boy Scout Rule" (leave the code cleaner than you found it).
- **Structural Organization:** Instructions to organize code into clearly-defined modules, manage dependencies correctly, and reuse existing utilities rather than creating redundant logic.
- **Code-Level Simplification:** Directives to write simple, clear code by avoiding deep nesting, simplifying complex conditions, and keeping functions short and focused.
- **Naming and Documentation:** A strong focus on clear, descriptive names for variables and methods, as well as the creation and maintenance of comments and docstrings to provide context for future developers.

Finally, developers instructed the agent to produce code that was resilient, efficient, and well-tested. They expected the agent to proactively avoid common pitfalls (e.g., unsafe memory operations).

These rules about code quality communicate a clear developer preference that the efficiency gain from agentic coding should not come at the expense of the long-term health and coherence of the codebase.

### 3.2.3 Solve Problems Effectively. 
Beyond adhering to institutional standards and ensuring code quality, developers actively seek to instill effective problem-solving heuristics in the agent through agent rules. These rules represent a form of procedural knowledge, teaching the agent not just *what* to do, but *how* to approach its tasks like an experienced engineer in the following aspects:

First, the agent should gather and internalize project-specific information before making changes. This involves reading documentation and examining existing code to understand the project's architecture, tech stack, and design patterns.

Second, when approaching a large task, the agent should methodically break it down into small, manageable steps. Directives such as "one build configuration file at a time," "small, single-purpose PRs," and making the "smallest possible code change" were common among agent rules.

Third, developers mandated that the agent take responsibility for the quality of its own output before presenting it to them. The agent was commanded to perform a range of validation actions, including building the project, checking code style, and running tests.

Fourth, to prevent unintended side effects or scope creep, developers imposed strict constraints on the agent's actions. Using imperative phrases like "Only accept," "DO NOT engage," and "Leave... alone," developers created clear boundaries, demanding that the agent remain laser-focused on the specific task at hand.

Fifth, in contrast to the need for strict focus, some developers granted the agent a degree of autonomy to resolve ambiguity, especially in low-stakes situations. Rather than halting and asking for clarification, the agent was sometimes expected to use its reasoning capabilities to infer the developer's intent from the available context.

Finally, related to inferring intent from context, the agent was also expected to reason about the best course of action from examples in the codebase.

### 3.2.4 Collaborate with the Developer. 
Developer defined protocols, through agent rules, for a productive human-agent partnership, focusing on how the agent should communicate, manage uncertainty, formulate plans, and learn over time. Four key collaborative behaviors emerged from our analysis:

First, developers required communication that was precise, transparent, and efficient, prioritizing substantive information over conversational pleasantries. This technical communication style aimed to ensure the precision of information, maintain the developer's situational awareness, and facilitate exception management.

Second, the agent was expected to recognize the limits of its own knowledge and capabilities. Under conditions of ambiguity, high-stakes actions, or environmental blockers, it was instructed to pause and defer to the developer. Critically, developers also wanted these interruptions to be low-friction, often by having the agent provide suggested answers to make it easy for the developer to provide guidance.

Third, before executing complex tasks, developers required the agent to engage in collaborative planning. This involved creating a plan, evaluating the pros and cons of different approaches, and presenting recommendations for approval. In some cases, the agent was even expected to exercise critical thinking, challenge suboptimal initial requests, and propose better alternatives with clear justification.

Last, developers sought to overcome the stateless nature of LLMs by instructing the agent to learn from its mistakes and experiences. While many rules implicitly aimed for this through negative constraints (e.g., "never ever do . . . "), some developers attempted to arrange an explicit mechanism for knowledge retention, such as having the agent maintain a "lessons_learned.md" document.

The above collaborative protocols defined in agent rules suggest a sophisticated interplay of developer needs. First, the demands for planning, transparency, and seeking help are primarily risk-mitigation strategies. Developers are concerned about agents performing destructive or incorrect actions without oversight and use these human-in-the-loop checkpoints to maintain control.

Second, efficient communication and the ability to learn from feedback are aimed at making the agent easier to manage. Conversational fillers is perceived as noise that slows down human guidance, while having to repeat corrections is a source of frustration.

Last, beyond simply preventing errors, some rules aim to leverage the agent's reasoning capabilities. By encouraging the agent to critically examine requests, analyze trade-offs, and identify ambiguity, developers are prompting the agent to exercise its "agency," though often within specified limits.

## 4  Expert Software Developer Interviews

We triangulated our taxonomy of agent behavior from the previous exercise by conducting semi-structured interviews with experienced software developers from the same organization. We framed the conversation around values and mindsets that the participants considered critical to be successful software engineer in their organization. We then asked them to reflect on how junior developers could "grow" to acquire them and how AI agents measure against them today. This helps us identify attributes that were not present in the agent rule files due to pragmatic reasons like developers' perceived capabilities and limitations of current LLMs and helps us establish a more aspirational north-star for a highly effective coding partner.

### 4.1  Methods

We conducted semi-structured interviews with 15 experienced software engineers who frequently used AI agents for work from the same large technology company (see demographics information in Figure 1). We situated our interview protocol in areas of professional growth expected of junior developers as they become seasoned practitioners. This anchoring is an intentional response to the popular framing of AI agents in industry headlines that they could already act like junior software developers[2].

Each interview included two main segments: 1) a mindmapping exercise aimed at eliciting the participant's view of what constitutes core values and principles in software engineering, often drawing from their own professional journey, and 2) a discussion about how junior developers and AI agents would respectively measure up to these stated ideals. Two of the co-authors conducted thematic analysis on the interview transcripts, informed by the codebook developed from analyzing the agent rules.

### 4.2  Findings

We found that the expected areas of growth for junior developers are generally aligned with the expectations of (and hopes for) AI agents expressed in agent rules today. For example, participants emphasized the importance of prioritizing code quality over the speed of coding, and believed the key to achieve high code quality is to employ sound problem-solving approaches, such as thinking holistically and making incremental changes, as well as collaborating with others effectively, especially through design and code reviews.

---

[2]Goldman Sachs deploys AI software engineer Devin to transform tech roles. https://www.hrkatha.com/news/goldman-sachs-deploys-ai-software-engineer-devin-to-transform-tech-roles/

Nonetheless, a crucial difference is that junior developers are expected to take initiatives in their own growth through seeking help and asking questions, observing team workflows and norms, and learning from feedback through both formal reviews and informal discussions with colleagues. For instance, P1 shared a particular mindset he benefited when he was in earlier stages of his career:

*"I think one of the things I've learned is… always make sure that you actually understand the code that you're reading and the technical changes. And don't be afraid to ask other engineers or challenge why the code was written that way. Don't make assumptions that it's already correct or that it should be intuitive."* - P1

And P10 emphasized the tacit knowledge that needs to be acquired through interacting with team members:

*"There is some sort of team norm (about coding patterns) that needs to be learned over time or through talking to the senior people on the team."* - P10

In contrast, participants expected AI agents to perform as is, and any seeming improvements of capabilities over the baseline would require active guidance from the user through prompt and context engineering. Therefore, this metacognitive and self-improving capability sets apart human capabilities from AI's, and should be an important consideration when defining and benchmarking against AGI. In the rest of this section, we elaborate on each of the core values junior developers are expected to adopt as they grow, and how the expectations differ when directed towards junior developers and AI agents.

*4.2.1 Ensuring Code Quality.* This is the value that many of our participants considered to be paramount. Participants described "quality" code as code that is correct, readable, maintainable, and thoroughly tested. Participants think of code quality as the social practice of communicating and collaborating with "future engineers" (P1) like "someone who doesn't work at [Company Name] yet" (P9). In that sense, writing high-quality code—through its readability, comments, and documentation—is a form of asynchronous communication. Tests were described as a "second tier of documentation" (P4) for future engineers.

*"I would say if we don't pay attention to the quality over time, it's gonna bite you back. ... because people come and go, you know, ... some people may have this knowledge and they're gone and we just totally lose that piece. And when we run into something like that, it would take much longer for a team member to figure that out."* - P7

Participants did not anticipate junior developers would appreciate the importance of code quality immediately. They acknowledged that this is an area of initial struggle and eventual growth. Participants expected junior engineers to internalize the mindset of quality first through mentorship, typically iterating on their code via the code review process, and overall building empathy for the reader of their code.

*"There is a tendency for earlier (in their career) developers to get the thing done, and that's it ... their first draft is usually their final draft. ... I think the first thing I try to, you know, work with them is 'let's iterate.' Do your own first code review, right? Take off your code author hat and put on a code reviewer hat."* - P6

On the other hand, an AI agent is still viewed as a tool, despite their potential to become a partner, to which tedious quality tasks

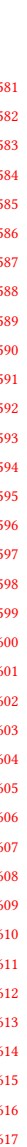
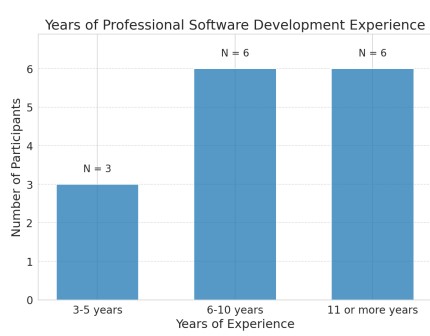
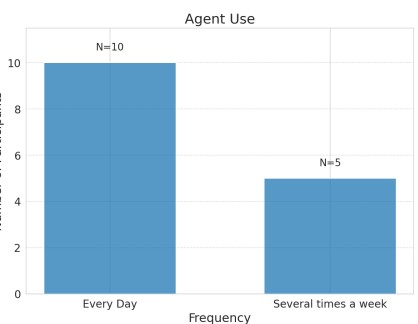
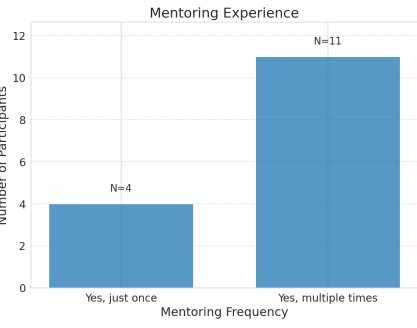

**Figure 1: The majority of the 15 developers in our interview study had more than 6 years of professional experience, and the majority use AI agents for their professional work every day. In addition, the majority of our participants supervised or mentored junior software developers multiple times in their respective careers.**

(like generating unit tests) can be offloaded immediately and at scale, with no learning curve required.

*"I think AI does a very good job of writing unit tests... normally if I had to write a unit test on my own completely, it'll take like a lot of time"* - P10

### 4.2.2 Adhering to Standards and Processes.

Closely related to code quality, participants viewed "internalizing" and adhering to team and organization-wide standards, both explicitly documented best practices as well as implicit "team habits" as an important trait. Adherence to standards enables order and predictability in a large, multi-person codebase, which in turn ensures that the codebase remains maintainable over time.

*"It's not just about the functionality, but also overall readability and whether it aligns with the rest of how [Company Name] writes this piece of code"* - P6

Much like in the case of code quality, participants viewed this as a trait that junior developers would acquire through learning from documentation and more importantly, acting on feedback from code reviews. Junior developers are expected to learn both the documented standards and unwritten, implicit ones embedded in the team's culture.

*"A lot of it is done in the (code) review process... My tech lead pointed me to the internal [Company Name] C++ documentation to follow like [Company Name] style practice."* - P1

In contrast, they expected AI agents to automate adherence to standards. For example, a coding agent who can rewrite any code to match the standards of surrounding code:

*"So what I did was I basically used <Agent Name> to say, rewrite what I've done, but in the style of the rest of this file."* - P9

Still, some participants were disappointed that AI agents were not yet able to learn what standards and rules should be enforced based on a project's context and an individual's past work. For example, P2 wanted an agent that can learn from his past code reviews and enforce similar standards in future review requests.

*"The AI could learn from this, [including] 1500 CLs (change lists) I reviewed and what comments I provided, and act on the new CLs."* - P2

### 4.2.3 Solving Problems Effectively.

Beyond writing good code, a key trait that participants identified as key to being an effective developer was to adopt the right methods for problem solving, including understanding both technical and business requirements, weighing tradeoffs, and considering long-term implications of a solution.

For human developers, participants noted that strategic insight emerges from accumulated experience and a holistic grasp of the system. Junior developers actively cultivate this comprehensive understanding through exploration, diligently "reading team documents," conducting "reasonable research" (P3, P4), and learning incrementally from working with the team over time.

*"Your level of understanding of the code base, right? kind of determines the code you can write... if you know more, like you have more context over larger code base, then you know how to design some part of it"* - P5

For AI agents, any strategic output is attributed to a masterfully crafted prompt. The intelligence and strategic insight originate with the human user; the AI functions merely as a tool to explore that strategy.

*"To me, AI is more of a tool, and so unless the user is thinking about these things first. I wouldn't trust AI to do all of this on behalf of the user yet."* - P14

Both AI and human developers share a common anti-pattern: an inclination to prematurely execute tasks. Junior developers often face criticism for adhering too rigidly to the problem described on a ticket, failing to grasp the "bigger picture" (P3). Similarly, AI agents exhibit a comparable shortcoming, executing prompts "without thinking about other consequences" (P5). In line with this sentiment, 90.11% of agent rule files contain rules related to the behavior "Understand Project Context before Acting". The critical distinction lies in how this narrowness is perceived: for junior developers, it represents a remediable developmental phase, whereas for AI, it is viewed as an inherent technical limitation.

*"...what would make me comfortable is if it (AI) could, um, I think making the code changes is the easier part. If it could also consider and show all the trade-offs it considered."* - P6

### 4.2.4 Collaborating with Others.

Effective collaboration and communication was perceived as an important trait to reduce misalignment and thus wasted work. Soliciting others' feedback early in one's problem solving process is considered essential to teamwork:

*"Code reviews are sort of, that can be too late. ...somebody who has already spent a bunch of effort on a design. And at that point, it's hard to provide input necessarily into some of the choices that are made ... you're going backwards."* - P12

For junior developers, effective communication was viewed as a developmental journey, requiring them to overcome a reluctance to seek assistance and internalize the understanding that software development is a collaborative endeavor. The expectation was that they should proactively communicate both progress and impediments, avoiding work "in a hole for three days" (P6), echoing the expectation that agents should also operate incrementally and iteratively. In addition, junior developers were expected to demonstrate effective communication by adding adequate context in CL descriptions to aid reviewers, as well as create more durable artifacts like documentation to enable asynchronous communication.

For AI Agents, proactive communication was required to transform its "black box" (P14) nature and build trust. Participants expected that AI agents need to be explicitly designed for interactive dialog and transparent communication. They expect agents to explain the reasoning behind suggestions and changes as well as proactively identify knowledge gaps and ask clarifying questions, even presenting tradeoffs.

*"I would expect more of a conversation-like [interaction]–here's the problem, here's [how] I would test it... but there was no such conversation [with AI]."* - P1

*"It needs to be able to figure out when it doesn't have enough information and ask... It's gonna be more of an interaction versus like correction... I expected it to say something. 'Oh, like I have these two possible scenarios... these are the trade offs for left and right... what would you recommend?'"* - P3

*4.2.5 Learn from Feedback and Past Experiences.* Closely related to being an effective collaborator, is being able to learn from feedback and past experiences. This proactive effort on part of the junior developer or AI greatly improves continuity and efficiency: the senior developer doesn't have to explain the context again, provide the same feedback again, or fix the same bugs as before.

Effective collaboration is intrinsically linked to the ability to learn from feedback and prior interactions. Exhibiting this behavior, whether by a junior developer or an AI agent, enhances efficiency and outcomes of teamwork, alleviating the need for senior developers to repeatedly re-explain context, reiterate feedback, or re-address recurring issues.

For junior developers, participants perceived learning from feedback as an active engagement with the code review process, necessitating an openness to constructive criticism. For Agents, learning from feedback was closely related to retaining context and personalizing suggestions based on a user's past actions.

Participants in the study acknowledged the potential for AI to be more than a just a tool and an effective partner, given it has already been trained on the organization's code base, but lamented that AI's stateless manner, as in its inability to remember the "conversation leading up to" the current moment and learn from past interactions, could make it a forgettable partner.

*"...I would make changes here just fixing bugs. Then if I added any other prompt. It would reproduce the old bug as opposed to taking into account the fact that I changed the code."* - P13

## 5 Discussions

### 5.1 The Evolving Mental Model and Expectations of AI Agents

Our work highlights an evolving mental model of AI agents in software engineering: the simultaneous perception of AI as both a powerful tool and a potential collaborative partner. This ongoing shift in turn influences what users expect from agents.

This transition of developer mental models is reflected in the varying abstraction levels of agent rules in the corpus we analyzed. Developers specified both high-level principles (e.g., *"Write for Others"*) as well as highly specific, step-by-step directives (e.g., *"After creating your new* `_test.go` *file, add a* `go_test` *target to the* `test/<package>` *file."*). The former suggests a desire for a partner capable of autonomous reasoning and adherence to broad guidelines, while the latter treats the agent as a tool that requires precise instructions to execute tasks. This duality underscores the current stage of AI development, where agents demonstrate advanced inference capabilities but still require explicit, step-by-step guidance in certain contexts.

Our interviews with senior developers, on the other hand, suggest the desirable state where this transition of AI from tool to partner should target. Notably, developers expect their teammates and work partners to be able to learn proactively and adjust their approaches based on past experiences–an emerging expectation we started seeing expressed towards AI agents as well in our rule analysis.

As AI models and agents advance, we anticipate this evolving mental model to be reflected in the users' expectations of agents, and in turn, in the agent rules. This necessitates the system and infrastructure to facilitate periodic updates of the behavior taxonomy. This must be coupled with a dynamic evaluation framework that accounts for shifting benchmarks and user needs.

### 5.2 Implications for Agent Evaluations

To effectively evaluate the progression from AI-as-a-tool to AI-as-a-partner, an agent behavioral taxonomy grounded in real-world user preferences, like the one we presented in the paper can serve both as a foundation and goalpost. It offers several ways to enhance our toolbox of evaluating AI, enabling more effective bi-directional human-AI alignment.

**Promote Benchmarks Focusing on Developer-AI Collaboration** Shifting the focal point of coding agent evaluation away from functional correctness towards collaboration and teamwork, our taxonomy can play a key role in bringing emerging AI evaluation and alignment techniques, such as measuring grounding in human-AI communication [24] and human-AI collaborative effort [23], into the mainstream of benchmarking LLM's software engineering capabilities, and thus realign the goalpost for AI labs, agent builders, and enterprises looking to further integrate AI agents into software development life cycles.

**Enable Qualitative Assessment of Agent Behavior**: Beyond benchmarking, a behavioral taxonomy enables multi-faceted qualitative assessment of agent performance. By leveraging techniques like "Report Cards" [29], subject-matter experts or well-calibrated LLM-as-a-Judge can assess agent trajectories and code output against

specific attributes in the taxonomy to contextualize traditional metrics such as *pass@k* with observational commentaries.

**Support Tailored User Feedback**: As a synthesis of user preferences, a taxonomy of expected agent behaviors can make user feedback targeted and actionable. Rather than presenting a pair of generic thumbs-up and thumbs-down buttons at the end of each chat session, we envision an adaptive feedback UI that invites the user to rate and comment on specific behavioral attributes most relevant to the session in question.

### 5.3 Implications for Agent Design

Raising the bar for evaluation to include traits like collaboration and learning, naturally introduces a positive pressure on evolving agent design to meet the new standards. Our results suggest a few key areas of improvement to foster more effective human-AI partnerships.

**System Instructions**: Our behavioral taxonomy provides a structured vocabulary for expressing common values in software engineering into concrete directives in the agent's system instructions. For instance, to uphold the value of *Code Quality*, a system instruction designer can consider different attributes of agent behavior within that expectation and then operationalize each into specific guidelines, such as instructing the agent to prioritize readability and avoid smart but hard-to-understand tricks.

Similarly, to foster better *Collaboration*, system instructions can require the agent to critically examine implicit assumptions in the user's request and the initial plans the agent comes up with. This approach transforms system instructions from functional configurations into living artifacts of a team's engineering culture.

**Memory and Personalization**: A significant limitation of current AI agents, as observed in our studies, is their limitation in maintaining state across sessions. This hinders their ability to learn from past interactions and adapt to individual user preferences, leading to frustration and the need for repeated corrections. Developing robust mechanisms for persistent memory and personalized learning will be essential for agents to evolve into truly adaptive and trustworthy partners.

**"Over-the-Shoulder" Learning**: To match the self-improving capacity of junior developers, as Study 2 suggests, AI agents need to learn from observing how their human partners perform tasks, in a manner similar to "over-the-shoulder learning"[26]. This involves developing mechanisms for agents to observe human workflows, identify patterns, and internalize best practices, thereby acquiring tacit knowledge that is often difficult to codify in explicit rules.

### 5.4 Implications for Foundational Models

At the core of SWE agents lies the large language models that drive them. However, because current models are typically optimized for short-term, next-turn rewards, they often struggle with the long-horizon dynamics of real-world collaboration. To overcome this bottleneck, the field must pivot toward training paradigms that prioritize sustained interaction—a shift already pioneered by works like CollabLLM [28]. Crucially, this evolution requires a corresponding shift in evaluation frameworks, as evaluation metrics heavily influence the trajectory of research and development.

### 5.5 Implications for Human Supervision

Finally, supporting users who supervise AI agents is a critical, often overlooked, aspect of fostering effective human-AI collaboration. This includes developing:

**Agent Rule Validators**: Tools that help users iteratively test and refine their agent rules against their own use cases can be considered as project-specific agent evaluations. This capability enables users to fine-tune agent behavior for optimal performance within their unique workflows and project requirements, ensuring that agents act as effective and reliable partners.

**Agent Rule Discovery and Adaptation**: Enabling agent rule discovery can empower users to learn successful strategies from their peers and subsequently adapt validated rules to their unique contexts. This fosters a community of practice for agent customization.

**Agent Profile**: Providing users with clear "agent profiles" that detail an agent's evaluated behavioral strengths and weaknesses can help users calibrate their trust and focus their rule authoring effort on specific types of agent behavior that require explicit guidance

### 5.6 Limitations

We acknowledge the following limitations in our research. First, the data for both agent rule analysis and developer interviews were collected exclusively from a single large technology company. While this allows for in-depth analysis within a specific context and triangulation of results, this choice may limit the generalizability of our findings to other organizational settings, company cultures, or industries. Second, the adoption of agent rules is rapidly growing, which could lead to changes and variations in their usage over time. Thus, future research could benefit from a broader corpus of agent rules to validate and expand upon our observations.

### 6 Conclusions

To sum up, our main contribution is a taxonomy of desirable AI agent behavior for enterprise software engineering, derived from a qualitative analysis of 91 sets of agent rules and interviews with 15 experienced professional software developers. The taxonomy defines four key expectations of agent behavior: *Adhere to Standards and Processes*, *Ensure Code Quality and Reliability*, *Solve Problems Effectively*, and *Collaborate with the Developer*, providing a systematic vocabulary for setting goals on effective human-AI partnership. This work is a step towards addressing a critical evaluation gap in the evolving landscape of AI agents for software engineering. Our approach to developing a taxonomy of expected AI agent behaviors bears general implications for aligning AI development with user preferences in domains not limited to software engineering.

### 7 GenAI Usage Disclosure

Throughout this paper, GenAI tools were utilized for copy editing. Additionally, these tools assisted with data analysis, as detailed in the pertinent study sections. The authors have reviewed the GenAI output for accuracy and deem any potential errors acceptable given the qualitative nature of our analyses.

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
