# OpenReview forum: "Towards AI as a Collaborative Partner: A Taxonomy of AI Agent Behavior in Software Engineering"
_ACM.org/AIWare/2026/Conference — AIware 2026_

### Official Review · Reviewer_QEDF · 2026-02-23

**Rating:** 2
**Confidence:** 5

**Review:**

(+) Strength

+ The study taxonomy is grounded in practitioner-defined rules and further validated with professional developers, which strengthens its practical relevance.

(-) Weaknesses

- The validation interviews seem confirmatory rather than exploratory. It is not fully clear whether participants challenged, refined, or rejected elements of the taxonomy.

- The study does not clarify whether saturation was reached during interviews, which makes it difficult to assess whether only 15 participants were sufficient.

- The study does not provide a replication package.

(*) Detailed Comment:

- The introduction is somewhat confusing and hard to follow, and does not clearly articulate the motivation for conducting the study.

- The main contribution of the work is not explicitly stated and could be clarified earlier in the paper.

- The authors do not provide a replication package with the materials used in the study.

- The related work section is difficult to follow and does not flow smoothly with the rest of the paper. I would recommend restructuring the section to cover the specific topics addressed by the study.

- Regarding the interviews, all 15 participants were recruited from the same company, which limits the diversity of perspectives and may affect generalizability. The sampling strategy and its implications for external validity are not sufficiently discussed. Also, explain if the participants were on the same team or were also the authors of the agent files analyzed?

- The interview protocol lacks further details, as the study does not clearly explain how questions were designed, whether they were piloted, how long interviews lasted, or how consistency was maintained across sessions.

- The mind-mapping exercise is not described in sufficient detail, as it is unclear how it was conducted, what prompts were used, how responses were captured, and how these artifacts were analyzed.

- The thematic analysis process also lacks transparency, it is not specified whether coding was performed independently, how disagreements were resolved, or whether inter-rater agreement was assessed.

- As the analysis was informed by a pre-existing codebook derived from agent rules, there is a risk of confirmation bias and the study does not clarify how new themes were allowed to emerge beyond the initial framework.

- The discussion section reads as an extension of the results, the section would benefit from deeper engagement with prior work and a clearer positioning of the findings within existing literature.

- The limitations section is very brief and should be expanded to address additional methodological and contextual constraints.

- Minor:

- Check if all acronyms (e.g., SWE, LLM) are defined at first use in the main text.

- Fix all checkmarks.

- Page 7, line 798 include “:” after Collaboration to keep the consistency.

**Summary:**

The study proposes a taxonomy of desirable behaviors for enterprise SWE agents, derived from 91 developer-defined rule sets and validated with 15 professionals.

---

> ### Author Response · Authors · 2026-03-18
>
> Dear Reviewer QEDF,
>
> Thank you for your thorough review and detailed feedback. We address your core concerns below:
>
> * **Exploratory vs. Confirmatory Interviews:** To minimize confirmation bias, we deliberately did not ask interview participants to comment directly on the taxonomy. Instead, the sessions were exploratory, focusing on understanding the core values expected of software engineers and how junior developers demonstrate them. We subsequently assessed the alignment between our derived taxonomy and the engineering values elicited during these interviews.
> * **Interview Saturation:** We will clarify in the text that we indeed reached saturation for the primary topic regarding the expected values in professional software engineering. However, participant views on the emerging capabilities of AI agents differed, especially concerning their self-improvement and continual learning capabilities.
> * **Replication Package:** Because the agent rule files are embedded within the company's proprietary codebase, we are unable to release the raw data. The same policy constraints apply to the interview transcripts. However, to support replication efforts on other comparable datasets, we will release the codebook and the prompt used for the LLM annotator as supplemental material.
> * **Participant Sampling and Overlap:** We will clarify our sampling strategy in the revision. All participants were drawn from different teams, and none of them were on the same team as any of the paper's authors.
> * **Methodological Details and Presentation:** We appreciate your detailed notes regarding our methodology. While we face space constraints, we will look for ways to make these details clearer in the revised manuscript. Specifically, we will note that the thematic coding was performed independently, and any disagreements were resolved through discussion. Furthermore, the mind-mapping exercise was used purely as an elicitation aid during the sessions; the resulting artifacts are not the primary data of the interview study. Finally, we will correct the minor typos, ensure all acronyms are defined at first use, and fix the formatting as noted.

---

> > ### Comment · Reviewer_QEDF · 2026-03-19
> >
> > Dear Authors,
> >
> > Thank you for the detailed and thoughtful response. I appreciate the clarifications provided.
> >
> > That said, some concerns remain. While you explain that the interviews were exploratory, the study still appears closely tied to the pre-existing taxonomy. Also, although you mention that saturation was reached, this is not clearly demonstrated in the paper.
> >
> > More broadly, the study would benefit from greater methodological transparency and reproducibility. Even considering constraints related to proprietary data, key aspects of the design (e.g., interview protocol, thematic analysis process, and data handling) are not described in sufficient detail to fully assess the rigor of the work. The absence of a full replication package, along with a limited explanation of how themes were derived and validated, makes it harder to evaluate the robustness of the findings. If possible, could you share a replication package at this stage? Overall, while the topic is relevant and promising, the current manuscript would benefit from stronger methodological clarity.

---

> ### Author Response · Authors · 2026-03-20
> **Re: Replication Package**
>
> Dear Reviewer,
>
> Thank you for your engagement in this discussion. Per your request, we have made a replication package available anonymously at [this URL](https://drive.google.com/file/d/1o7L1J9ieFN-G6b0U1AGV1QZgtzzKWgFL/view?usp=sharing). While we are not able to release our data, the package includes the following items used in the research process:
>
> - Prompt for the LLM-based Agent Rule Annotator
> - Codebook for Qualitative Analysis of Agent Rules
> - Discussion Guide for Developer Interviews
> - Codebook for the Interview Study
> - Participant Screener for the Interview Study
>
> We'd like to further clarify the relationship between the interview study and the agent rules analysis. We started the interview study a few weeks after we commenced the rules analysis project but before the results of the latter was solidified. The final codebook for the interview study, included in the replication package, was therefore influenced by the themes emerged from the rules analysis for the purpose of making direct comparisons between expectations expressed towards AI agents and those towards human software developers.
>
> I hope the replication package is helpful, and please let us know if there is anything else we can further clarify.
>
> Best regards,
>
> Authors

---

> > ### Comment · Reviewer_QEDF · 2026-03-20
> >
> > Dear Authors,
> >
> > Thank you for sharing the replication package.
> >
> > I appreciate the effort to make additional materials available despite the constraints around proprietary data. The inclusion of the interview guide, codebooks, and prompts is helpful and improves the study's transparency.

---

### Official Review · Reviewer_dXJw · 2026-02-26

**Rating:** 3
**Confidence:** 4

**Review:**

### Strengths
1. Timeliness: The paper targets a practical and emerging SE problem: how coding agents can collaborate effectively with developers, making the study valuable for both research and industry practice.
2. Validity: The taxonomy is grounded in real enterprise artifacts and then triangulated with interviews, strengthening the confidence that the findings reflect real-world workflows.
3. Clear structure: The combination of qualitative content analysis and semi-structured interviews provides a coherent narrative.

### Weaknesses
1. Figure readability: The text and labels in some figures are too small in the current formatting.
2. Insufficient ethics/compliance details: The interview study lacks explicit reporting on IRB/ethical review (or exemption), consent procedures, anonymization, and data handling, which reduces transparency.
3. Generalizability discussion: Since interviews come from one company, the paper could more explicitly discuss how organizational context might shape expectations and the potential limitations in generalization.

### Comments
1. Readability of Figures: The text and labels in Figure 1 are generally too small, making reading difficult within the paper's format. It is recommended to increase the font size or adjust the layout to improve readability.
2. Research Ethics and Compliance Information: Section 4.1 describes semi-structured interviews with 15 experienced engineers from the same large technology company, but the current description is not clear enough about IRB/ethical review, informed consent from participants, anonymization, and data processing methods. It is recommended that the authors add details such as whether IRB or equivalent review/exemption was conducted, how consent was obtained, how data was de-identified and stored/shared, and how potential conflicts of interest were handled, etc., to enhance the traceability and credibility of the research.
3. Paper Scope: The authors could add a short paragraph at the end of the `Related Work` section to clarify the differences between this paper and existing work, helping readers quickly grasp the incremental and unique aspects of this paper compared to related studies.
4. The interview participants are from the same company, which affects the generalizability of the outcomes. The authors should at least discuss this limitation in the paper.

**Summary:**

This paper focuses on the question of `how to transform AI coding agents from tools into collaborative partners`, which is a problem with detailed research and application value in real-world development.
Based on qualitative content analysis of agent rules written by developers for coding agents in enterprise environments, the authors summarize an AI agent behavior taxonomy centered on developer preferences.
This is further validated and supplemented through semi-structured interviews with senior engineers, ultimately extracting four core expectations and providing a framework for future agent evaluation and design in real-world engineering collaboration scenarios.

Overall, this paper has a novel and important topic, its methods and materials closely resemble the context of real-world enterprise software engineering, and the resulting taxonomy is inspiring for building evaluation and interaction design.
However, there are still some points for improvement in its presentation.

---

> ### Author Response · Authors · 2026-03-18
>
> Dear Reviewer dXJw,
>
> Thank you for your supportive review and for acknowledging the validity and timeliness of our research. We address your suggestions for improvement below:
>
> * **Figure readability:** We appreciate the feedback on Figure 1 and will ensure the text and labels are adjusted to be fully legible in the camera-ready version.
> * **Research Ethics and Compliance:** We will add the missing compliance details to the manuscript. To clarify, the authors are employees of the company where the research was conducted. While the company does not have a formal Institutional Review Board (IRB), the interview study was approved by the company’s legal department, and we obtained informed consent from each participant prior to their interview session.
> * **Generalizability and organizational context:** We will explicitly discuss the limitations regarding generalizability. We will clarify that the organization studied maintains a decades-old codebase with tens of thousands of active software engineers. Our findings reflect the maturity of the company’s codebase and the highly collaborative nature of software development practiced within this enterprise environment.
> * **Related Work Scope:** We appreciate your suggestion. To clarify, the paragraphs spanning lines 160–165 and lines 187–191 are both intended to explain how our study builds upon prior work. In our revision, we will revisit these specific sections to make them clearer and distinctly highlight the incremental and unique contributions of our study compared to existing literature.

---

> > ### Comment · Reviewer_dXJw · 2026-03-20
> >
> > Thanks for the authors' timely response. The author has clarified most of the comments and problems in my review, which addressed my concerns to some extent.
> > For the `research ethics and compliance` part, it would have been even better if the response could provide more specific details (e.g., anonymization and data processing).
> > I decide to keep the positive rating.

---

### Official Review · Reviewer_3u5r · 2026-03-11

**Rating:** 3
**Confidence:** 3

**Review:**

## Significance

Strength:
- The problem being studied is relevant and very timely. As the capability of SWE agents reaching a level where they can be widely used, there is a need to study the socio-technical aspect of SWE agents for them to contribute more effectively in software engineering teams.
- A clearly presented taxonomy can help derive benchmarks on the relevant aspects, and in turn drive the agent development.

Weakness:
- The capability of agents are changing fast. Both agent rule files and interviews capture what engineers expect from agents at one point of time. Since their capability may increase very rapidly, some findings may be less relevant in the future and another round of study would need to be performed. For example, the interview suggests that AI has “an inclination to prematurely execute tasks” (line 676). However, as of today, features like “plan mode” has been added to AI agents for them to autonomously draft a comprehensive plan before executing the task.


## Originality

Using agent rule files to derive an initial taxonomy and further validating them through interviews with software engineers is novel, and also adds rigor to the study.


## Clarity

Strength:
- The taxonomy is clearly presented in Table 1. The examples also help with understanding of what each behavior refers to.
- The quotes from developers further add context and clarity.

Weakness:
- One of the motivations of the paper is to encourage the development of new benchmarks on SWE agent behaviors beyond functional correctness. However, Section 5.2 mostly discusses the implication for agent evals on a high level, without more specific discussions on what aspects are most the important for benchmarking. It would be better to include more actionable items.
- There is partial overlap between “Adhere to Standards and Process” and “Ensure Code Quality and Reliability”. Both of them include the behavior of following existing guidelines like a certain coding style.
- There are some contradicting points between the taxonomy based on agent rule files and the interview. For example, the interview suggests that AI agents are expected to perform as is. However, Table 1 contains the expected behavior of “Learn from Feedback and Past Experiences”. It would be better to have more comprehensive discussions about the contradicting points.


---

Overall, the paper addresses an important problem through a rigorous process of synthesizing from agent rule files and validating through interviews. The paper can be further improved by including more actionable directions in developing new benchmarks based on the derived taxonomy and interview results.

**Summary:**

This paper presents a taxonomy of desirable SWE agent behaviors beyond functional correctness. The taxonomy focuses on aspects that make a SWE agent an effective contributor within a software development team. The taxonomy is derived by synthesizing from 91 agent rule files (such as CLAUDE.md) for SWE agents and is then validated by interviewing professional software engineers. Four core expectations including Adhere to Standards and Process, Ensure Code Quality and Reliability, Solve Problems Effectively, and Collaborate with the Developer are developed, together with specific behaviors from each expectation.

---

> ### Author Response · Authors · 2026-03-18
>
> Dear Reviewer 3u5r,
>
> Thank you for your constructive feedback and for recognizing the timeliness and relevance of our work. We address your specific concerns below:
>
> * **Relevance amidst fast-changing capabilities:**  Our research contributes a reusable methodology for synthesizing user preferences regarding AI in software engineering, which remains valuable as capabilities evolve. In addition, we believe the industry’s ongoing effort to address these exact behavioral attributes validates our taxonomy. Furthermore, our agent behavior taxonomy remains highly relevant, especially the behavioral attributes about human-AI collaboration, given that the most popular coding benchmarks (e.g., variants of SWE-bench and more recently Terminal-Bench) still focus on functional correctness.
> * **Actionable items for agent evaluations (Section 5.2):** The proposed taxonomy can directly guide the development of rubric-based LLM-as-a-judge techniques. In the revised manuscript, we will include specific examples of this, such as recent work on the topic (e.g., [https://arxiv.org/abs/2601.04171](https://arxiv.org/abs/2601.04171), and [https://arxiv.org/abs/2603.03800](https://arxiv.org/abs/2603.03800)).
> * **Overlap between Core Expectations 1 and 2:**
>   * Yes, there is partial overlap when it comes to the topic of code quality. While the  "Adhere to Standards and Processes" category emphasizes the process the agent follows and the documentation it consults in the course of performing a development task, the latter is focused on the resulting outcome.We made Code Quality a separate expectation of agents, because the topic is highly prominent in both the corpus of agent rules we analyzed and our developer interviews.
> * **Contradicting points on agent self-improvement:** The divergence between the agent rule files (learning from feedback) and the interview findings (performing as-is) highlights that there is currently no consensus on expectations for agent self-improvement. We will expand our discussion to clarify that these are emerging, and sometimes competing, views that accurately represent the active transition of AI from a tool to a collaborative partner.

---

> > ### Comment · Reviewer_3u5r · 2026-03-21
> >
> > Dear Authors,
> >
> > Thank you for the detailed response, which addresses most of my concerns. I appreciate the addition of discussion on actionable items and agent self-improvement.